# Transport mechanism of human bilirubin transporter ABCC2 tuned by the inter-module regulatory domain

Yao-Xu Mao[1,2,3,4], Zhi-Peng Chen[1,2,4], Liang Wang[1,2], Jie Wang [1,2],
Cong-Zhao Zhou [2,3] ✉, Wen-Tao Hou [1,2,3] ✉ & Yuxing Chen [1,2,3] ✉

Bilirubin is mainly generated from the breakdown of heme when red blood cells reach the end of their lifespan. Accumulation of bilirubin in human body usually leads to various disorders, including jaundice and liver disease. Bilirubin is conjugated in hepatocytes and excreted to bile duct via the ATP-binding cassette transporter ABCC2, dysfunction of which would lead to Dubin-Johnson syndrome. Here we determine the structures of ABCC2 in the apo, substrate-bound and ATP/ADP-bound forms using the cryo-electron microscopy, exhibiting a full transporter with a regulatory (R) domain inserted between the two half modules. Combined with substrate-stimulated ATPase and transport activity assays, structural analysis enables us to figure out transport cycle of ABCC2 with the R domain adopting various conformations. At the rest state, the R domain binding to the translocation cavity functions as an affinity filter that allows the substrates of high affinity to be transported in priority. Upon substrate binding, the R domain is expelled from the cavity and docks to the lateral of transmembrane domain following ATP hydrolysis. Our findings provide structural insights into a transport mechanism of ABC transporters finely tuned by the R domain.

Bilirubin is mainly generated from the breakdown of heme, the catabolism of which is an indispensable process that cleans the aged or abnormal red blood cells[1]. In hepatocytes, the highly hydrophobic bilirubin molecule is subject to conjugation with two glucuronide groups by the UDP-glucuronosyltransferase, producing the hydrophilic conjugated form[2], which is secreted to the bile duct by an ATP-binding cassette (ABC) transporter ABCC2[3], and eventually released into the small intestine and eliminated in feces[4]. Accumulation of excessive bilirubin in the cell will cause jaundice, liver disease or gallbladder dysfunction[5], especially neonatal jaundice[6] and brain damage[7] to infants. As a predominant conjugated bilirubin transporter, dysfunction of ABCC2 usually accompanies with bilirubin accumulation in

the liver that causes the hereditary conjugated hyperbilirubinemia[8], also known as Dubin-Johnson syndrome[9]. In addition, ABCC2 could also export excess estradiol to the bile duct in the conjugated form estradiol-17β-D-glucuronide ($E_2$17βG)[10]. It has been established that $E_2$17βG trans-inhibits the function of bile salt export pump (BSEP, also known as ABCB11) in a dose-dependent manner after secreted into bile canaliculi by ABCC2[11]. A study in rats has demonstrated that ABCC2 is essential for $E_2$17βG to induce cholestasis[12], suggesting a possible relationship between ABCC2 and human intrahepatic cholestasis of pregnancy[13].

ABCC family consists of 12 ABC proteins[14], which are divided into three groups: nine multidrug-resistance proteins (MRPs), including

[1]Department of Endocrinology, Institute of Endocrine and Metabolic Diseases, The First Affiliated Hospital of USTC, and Center for Advanced Interdisciplinary Science and Biomedicine of IHM, Division of Life Sciences and Medicine, University of Science and Technology of China, Hefei, Anhui 230027, China. [2]School of Life Sciences, Division of Life Sciences and Medicine, University of Science and Technology of China, Hefei, Anhui 230027, China. [3]Biomedical Sciences and Health Laboratory of Anhui Province, University of Science and Technology of China, Hefei, Anhui 230027, China. [4]These authors contributed equally: Yao-Xu Mao, Zhi-Peng Chen. ✉e-mail: zcz@ustc.edu.cn; todvince@mail.ustc.edu.cn; cyxing@ustc.edu.cn

ABCC2 (also known as MRP2), two sulfonylurea receptors (SUR1/ABCC8 and SUR2/ABCC9) and an ion channel, the cystic fibrosis transmembrane conductance regulator (CFTR/ABCC7). As an MRP, ABCC2 possesses a broad spectrum substrate specificity, and exports a number of anticancer drugs such as vincristine, cisplatin, doxorubicin, methotrexate, irinotecan, and paclitaxel, hence gaining the attention for cancer therapy[15,16]. Moreover, ABCC2 also mediates the export of endogenous or exogenous organic anions conjugated with glutathione, sulfate or glucuronide[17], thus playing an important role in detoxification. Human ABCC2 is specifically expressed in the apical membrane of hepatocytes[18], renal proximal tubule cells[19], and enterocytes of duodenum[20] and jejunum[21]. ABCC2 is a 180 kDa full transporter of 1545 residues that structurally consists of two half modules, which are predicted to be the transmembrane domain (TMD) and a cytoplasmic nucleotide-binding domain (NBD), respectively. Notably, ABCC2 also has an extra N-terminal TMD0, which exists mainly in ABCC family and was suggested to participate in the subcellular localization[22,23].

In this work, we report structures of ABCC2 determined by single-particle cryogenic electron microscopy (cryo-EM) at three different states: the apo-form, the substrate-bound, and ATP/ADP-bound structures, respectively. These structures reveal that the R domain of ABCC2 undergoes drastic conformational changes through the transport cycle. Structural analysis combined with biochemical assays delineates the fine regulatory role of R domain on the transport activity of ABCC2. Our findings provide not only structural insights into the substrate specificity, but also a transport mechanism of ABC transporters in ABCC family finely tuned by the R domain. In addition, the structural and biochemical analysis enable us to interpret the molecular basis of pathogenesis of the Dubin-Johnson syndrome.

## Results

### Biochemical characterization and structure determination of ABCC2

The full-length human ABCC2 was recombinantly expressed in human embryonic kidney 293 F (HEK293F) cells and purified to homogeneity in detergent micelles (Supplementary Fig. 1a, b). The ABCC2 protein was extracted from the membrane with detergent n-dodecyl-β-D-maltoside (DDM) plus cholesteryl hemisuccinate (CHS) and exchanged to digitonin during purification. The purified wild type (WT) ABCC2 protein displayed an ATPase activity with $K_m$ and $V_{max}$ values of $0.20 \pm 0.02$ mM and $142.3 \pm 2.9$ nmol Pi min$^{-1}$ mg$^{-1}$ protein, respectively (Fig. 1a). ABCC2 has a degenerated catalytic site in NBD1, as the catalytic residue Glu is replaced by an Asp, hence its ATPase activity is only supported by Glu1462 at the consensus site (Supplementary Fig. 1c). We therefore generated a variant of ABCC2 by replacing the Glu1462 with Gln (E1462Q), which exhibited completely abolished ATP hydrolysis (Fig. 1a), as expected. In addition, the activity of ABCC2 could be stimulated upon addition of bilirubin ditaurate (BDT), an analog of conjugated bilirubin (Supplementary Fig. 1d), displaying a half maximal effective concentration (EC$_{50}$) value of $1.17 \pm 0.12$ μM and $V_{max}$ of $186.5 \pm 4.1$ nmol Pi min$^{-1}$ mg$^{-1}$ protein (Fig. 1b). The results of these biochemical assays indicated that our protein samples are in a physiologically relevant state.

To gain structural insights into ABCC2, we solved structures of ABCC2 at three different states (Fig. 1c): an apo-form structure at 3.6 Å

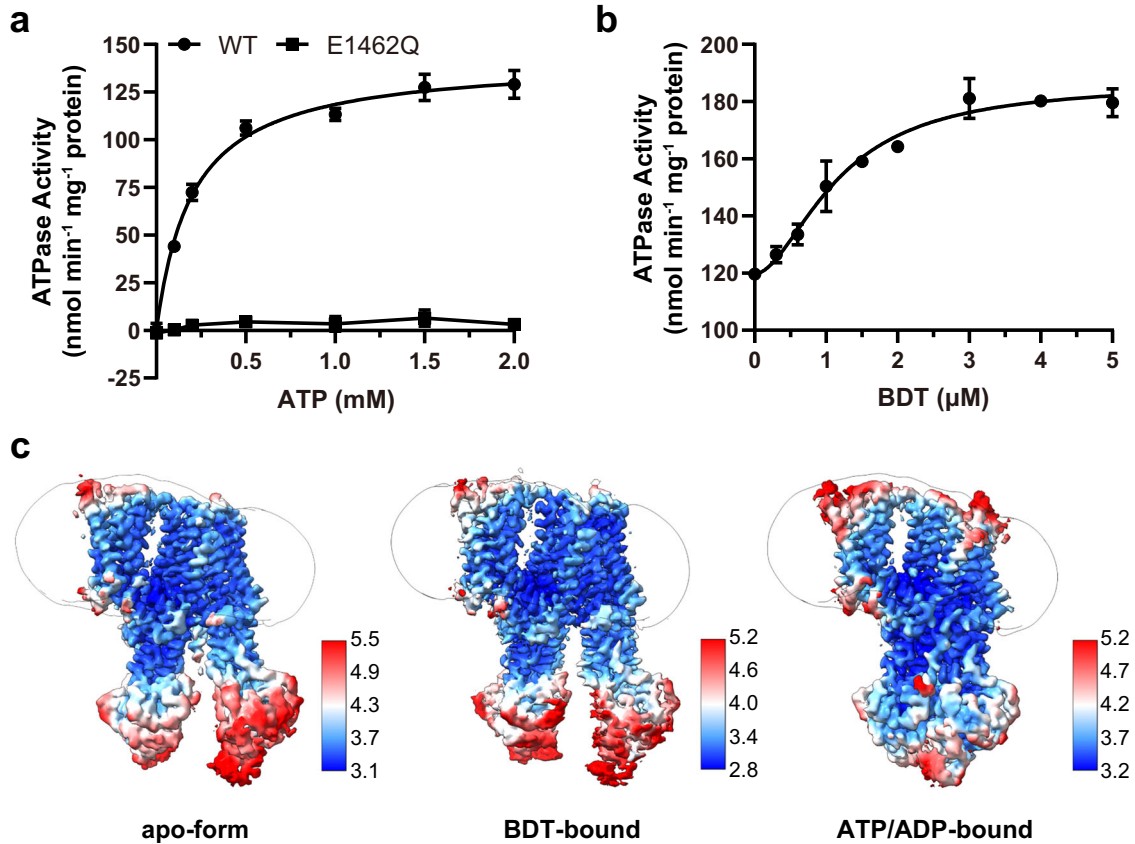

**Fig. 1 | Biochemical characterization and structure determination of ABCC2.**
**a** The ATPase activities of ABCC2 wild type (WT) and E1462Q mutant. The data points of WT activity are fitted with the Michaelis-Menten equation.
**b** The substrate-stimulated ATPase activity of ABCC2 upon the addition of conjugated bilirubin analog BDT. The data points are fitted with the Hill equation.

All data points of (**a**) and (**b**) represent means of independent experiments ($n = 3$) and the error bars indicate the means ± standard deviation (SD). **c** The refined cryo-EM maps of three ABCC2 structures. The unsharpened maps are displayed as the outline to show the position of detergent micelle. The cryo-EM maps are colored by UCSF ChimeraX 1.5 according to the local resolution estimated by cryoSPARC 3.1.

(Supplementary Figs. 2, 3), a BDT-bound structure at 3.3 Å (Supplementary Figs. 4, 5) and an ATP/ADP-bound structure at 3.6 Å (Supplementary Figs. 6, 7). Similar to most members of ABCB and ABCC family[24], ABCC2 also contains an insertion between two half modules. The counterpart insertion in CFTR has been named the R (regulatory) domain[25] and demonstrated as an activity regulator of the chloride channel through phosphorylation[26,27]. Thus, we also termed this inter-module insertion region (Lys857 to Lys961) the R domain. Notably, the R domain showed different configurations: its N-terminal segment (Lys857-Asp883, termed the N-segment hereafter) and middle segment (Asp884-Ser914, termed the M-segment hereafter) could be found in the translocation cavity in the apo-form structure, and only the C-terminal segment (Arg915-Lys961, termed the C-segment hereafter) was found at the lateral of TMD2 in the ATP/ADP-bound structure. In contrast, no density of the R domain could be found in the BDT-bound structure.

## The R domain occupies the translocation cavity in the apo-form structure

The apo-form structure of ABCC2 shows an inward-facing (IF) conformation (Fig. 2a, b). The NBDs possess a classic NBD fold of ABC transporter, with an α-helical subdomain and a RecA-like ATPase core subdomain. Unlike most typical type-IV exporter[28], ABCC2 possesses three TMDs (Fig. 2c). The extra TMD at the N-terminus of ABCC2 consists of five TMs, which is commonly named TMD0, and featured by most members of ABCC and ABCB families[29]. Several studies revealed the function of TMD0, such as SUR (ABCC8 and ABCC9)[30], TAP (ABCB2/3)[31] and TAPL/ABCB9[32]. TMD0 in ABCC family, for example, in SUR1/ABCC8 has been found to provide an interface to Kir6.2 tetramer which forms a central potassium ion channel[33]. On the other hand, in ABCB family, TMD0 of TAPL/ABCB9 is a module essential and sufficient for lysosomal trafficking[32], but dispensable for substrate transport[34]. However, the function of TMD0 in multidrug resistant ABC transports, including ABCC2, remains unclear. Each of the core TMDs of ABCC2 (TMD1 and TMD2) consists of six transmembrane helices (TMs) with a significant cytosolic extension forming two diverged wings at the hepatocellular side, and TMs packing against each other at the luminal side of the biliary canaliculus. TM9 and TM10 from TMD1 (or the counterpart TM15 and TM16 from TMD2) are swapped to the opposite TMD (Fig. 2a–c). Two pairs of coupling helices (CHs) from the TMD, which are embedded in the grooves on the NBD at the same side, couples the conformational changes between TMDs and NBDs (Fig. 2a–c). A linker that connects TMD0 and TMD1 has been found in all members of ABCC family, and commonly termed the lasso motif[28]. In our apo-form ABCC2 structure, the lasso motif is also folded into two helices (Fig. 2a, b).

In the translocation cavity, we found an extra density that could be fitted by the N- and M-segment of the R domain (Fig. 2a, b), in which only the side chains of the M-segment can be well fitted due to a relatively higher local resolution. The M-segment is folded into a short helix and a loop, occupying the translocation cavity, whereas the densities of the C-segment are untraceable in the map (Supplementary Fig. 3a).

The M-segment of the R domain is fixed in translocation cavity via abundant hydrogen bonds and salt bridges with the residues from TMD1 and TMD2. Specifically, Glu892 from the loop of the M-segment forms a hydrogen bond with Tyr381 from TMD1. In addition, Glu893 forms a hydrogen bond with Asn1204 and two salt bridges with Arg1205 and Arg1257 of TMD2, respectively. Arg904 from the helix forms a salt bridge with Glu1261 of TMD2 (Fig. 2d). To investigate the function of the M-segment that binds to the translocation cavity, we generated the variants E892Q and E893Q for ATPases activity assays. Interestingly, both variants displayed a significant increase of ATPase activity compared to the wild type (Fig. 2e). We further performed the transport activity assays using radioisotope-labeled substrate $E_2$17βG

and found that E892Q and E893Q also exhibited a significant increase of transport activity (Fig. 2f). Thus, these results indicated that the R domain binding to the translocation cavity functions as an auto-inhibitor of ABCC2.

## Upon substrate binding, the R domain is repelled from the translocation cavity

To decipher the binding pattern of conjugated bilirubin to ABCC2, we solved the structure of substrate-bound ABCC2 via addition of BDT (Fig. 3a, b and Supplementary Fig. 4). In this structure, the R domain is missing in the translocation cavity, which is alternatively occupied by a BDT molecule and an extra density (Supplementary Figs. 5, 8a). The extra density can be better fitted by a cholesterol molecule rather than CHS (Supplementary Fig. 8b), as the cholesterol is abundant in the cell membrane and may be co-purified with ABCC2. Thus, we tentatively fit this density with a cholesterol molecule. The two taurine moieties of BDT are stabilized by positively charged residues, namely Asn1204, Arg1205, Arg1257 from TMD2 and Lys329 from TMD1, respectively (Fig. 3c). The four pyrrole moieties are fixed by nonpolar residues together with the cholesterol molecule. Specifically, aromatic residues, namely Phe382, Phe437, Phe591 and Phe550 from TMD1, as well as Trp1254 from TMD2 provide ample π–π stacking with pyrroles to stabilize the BDT molecule in the substrate binding pocket.

Superposition of this BDT-bound structure with the apo form yields a root-mean-square-deviation (RMSD) of 0.783 Å over 1263 Cα atoms (Fig. 3d), indicating little conformational changes upon substrate binding. The superposition reveals that the binding site of BDT and cholesterol is largely overlapped with that of the R domain in the apo form (Fig. 3d). All key residues are almost aligned, except for a side-chain rotation of Trp1254, which is sandwiched between BDT and cholesterol (Fig. 3e). It suggested that the R domain and substrate share a competitive binding site.

Moreover, mutagenesis combined with ATPase activity assays revealed that ABCC2 variants with single mutation of the substrate-binding residues, either those forming polar interactions (K329A, N1204A, R1205A, R1257A) or hydrophobic interactions (F382A, F550A, F591A, W1254A), displayed a significant decrease of relative BDT-simulated ATPase activity (Fig. 3f), compared to the wild type. The only exception F437A showed a very sharp decrease of the basal ATPase activity (Supplementary Fig. 8c), thus yielding a higher relative BDT-stimulated activity (Fig. 3f).

## The R domain docks to the lateral of TMD2 in the ATP/ADP-bound ABCC2

To better understand the full transport cycle of ABCC2, we solved the ATP/ADP-bound ABCC2 structure at 3.6 Å (Fig. 4a, b and Supplementary Fig. 6). In the structure, two strong and well-defined densities were observed, which correspond to one ADP molecule at the consensus active site Glu1462, and one ATP-$Mg^{2+}$ molecule at the degenerated active site Asp786, respectively (Fig. 4a and Supplementary Fig. 8d, e). It represents a state that the ATP molecule at the consensus site has been hydrolyzed but remaining unreleased, which mimics the so-called active turnover state as captured in bABCC1[35,36]. Similar to bABCC1, the ATP/ADP-bound ABCC2 also adopts an outward-facing (OF) conformation, leading to the collapse of the binding pocket that accommodates BDT or the R domain (Fig. 4b, c and Supplementary Fig. 8f).

Remarkably, we could unambiguously identify the density of residues from Lys947 to Lys961 of the C-segment of the R domain, displaying as a continuous density linked to TMD2 (Fig. 4d). The C-segment extends from the elbow helix of TMD2 along the cleft between TM14 and TM15 in the cytosolic side, reaching the interface between the two NBDs. Residues from the C-segment forms a couple of hydrogen bonds and salt bridges with TM14, TM15 and NBD1: a

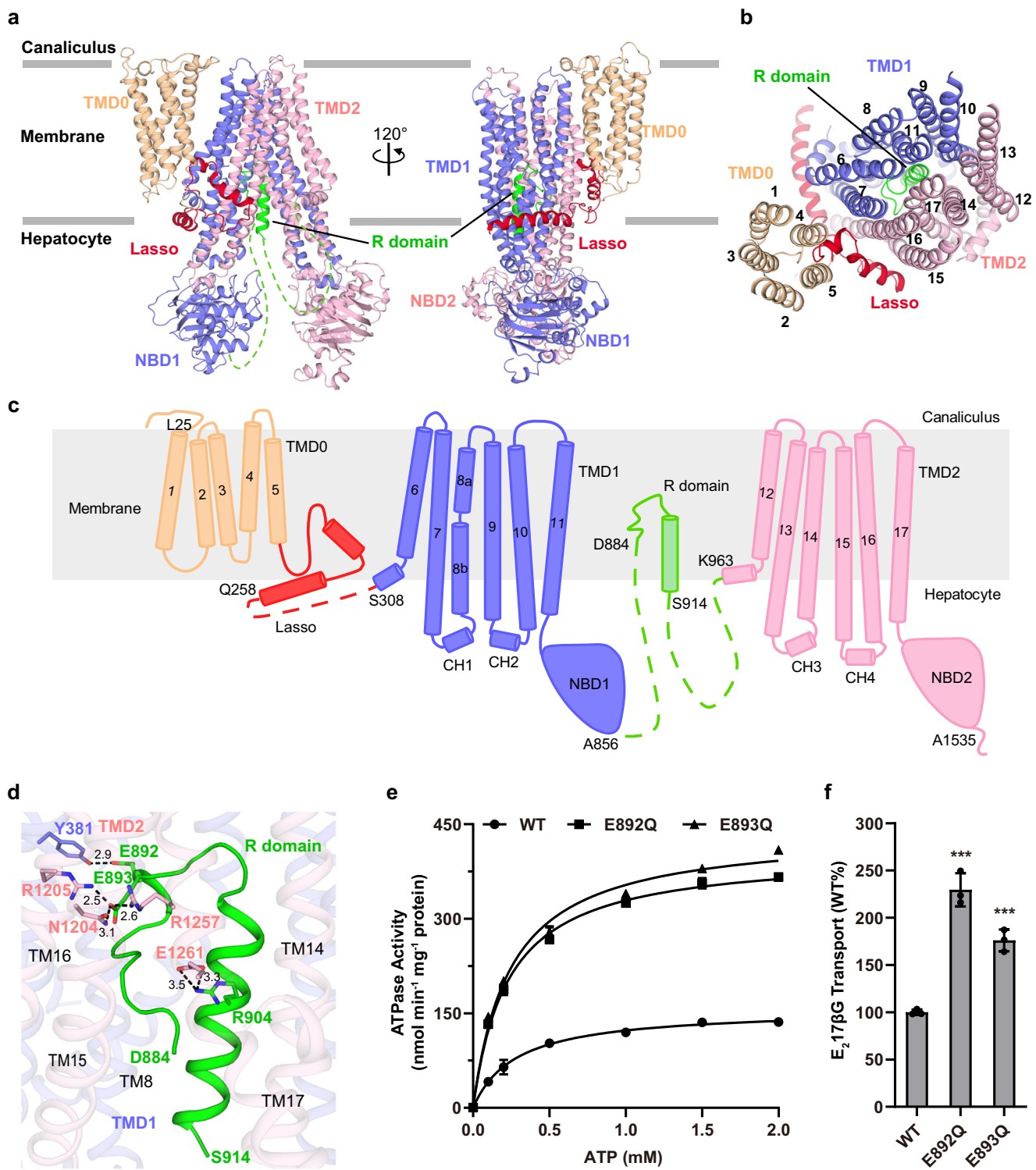

hydrogen bond between Gly960 and Arg1146 from TM15, a hydrogen bond between Glu958 and Arg1150 from TM15, a salt bridge between Glu955 and Arg1083 from TM14, and a hydrogen bond between Lys953 and Arg1079 from TM14. Meanwhile, Lys950 and Gln949 in the C-segment are further stabilized by two salt bridges and a hydrogen bond with Glu753 from NBD1. Sequence alignment revealed that these C-segment binding residues are also highly conserved in ABCC2 homologs (Supplementary Fig. 8g, h). In conclusion, in this active turnover state, the C-segment acts as staples that freezes ABCC2 at the OF conformation by crosslinking TM14 and TM15, even after ATP hydrolysis and substrate release.

Our present structures also provided the molecular insights into the etiopathology of clinical mutations in the *ABCC2* gene. To date, more than 70 mutations have been reported in the Human Gene Mutation Database (https://www.hgmd.cf.ac.uk/ac/index.php), in which 50 (involved in 13 residues) are Dubin-Johnson syndrome associated missense mutations (Supplementary Fig. 9). Among which, six mutations are found in NBDs, which might interfere ATP binding or hydrolysis of ABCC2, whereas seven mutations are mapped to TMDs, which might alter the process of substrate translocation. Of note, R1150H, which occurs at the lateral side of TMD2, has been reported to cause the serious Dubin-Johnson syndrome[37]. In our present structure

**Fig. 2 | Overall structure of the apo-form ABCC2. a** Cartoon representation of the apo-form ABCC2. The TMD1&NBD1 module is colored in slate, the TMD2&NBD2 module is colored in lightpink, TMD0 is colored in wheat, the lasso motif is colored in meitnerium and the R domain is colored in green, respectively. The unmodeled regions of the R domain are represented by dashed lines. The apical canalicular membrane of hepatocytes is indicated as the gray lines. **b** Top view of cartoon representation of the TMDs. The transmembrane helices (TMs) are sequentially numbered. **c** Topological diagram of ABCC2 is colored using the same color scheme as shown in (**a**). Terminal residues of each structural segments are labeled. The TMs and coupling helices (CH) are sequentially numbered. The membrane plane is indicated as the gray rectangle. **d** The interactions between the R domain and TMDs. Interacting residues are shown as sticks. Carbon atoms are colored consistent with domain colors in (**a**), with oxygen in red and nitrogen in blue. Hydrogen bonds and salt bridges are shown as black dotted lines, with the distance in Å. **e** The ATPase activities of ABCC2 WT and two mutants in the R domain. The data are fitted using the Michaelis-Menten equation. Each data point is the average of three independent experiments ($n = 3$), and error bars represent the means $\pm$ SD. **f** The transport activity assays of ABCC2 and two mutants in the R domain, using radioisotope-labeled substrate $E_2 17\beta G$. The transport activities of mutants are normalized by WT. Each data point is the average of independent experiments ($n = 3$), and error bars represent the means $\pm$ SD. One-way analysis of variance (One-way ANOVA) is used for the comparison of statistical significance. The $p$ value of E892Q is <0.0001, and the $p$ value of E893Q is 0.0007. The $P$ values of <0.05, 0.01, and 0.001 are indicated with *, **, and ***, respectively. Source data are provided as a Source Data file.

of ATP/ADP-bound form, this mutation disrupts the hydrogen bond between Arg1150 and Glu958, thus weakening the binding of the C-segment to TMD2. In fact, this R1150H variant showed a similar purification profile as the wild type (Supplementary Fig. 1b), while it displayed a significantly higher substrate-stimulated ATPase activity upon addition of either BDT or $E_2 17\beta G$, compared to the wild type (Fig. 4e, f). In contrast, the R1150H variant possesses $EC_{50}$ values of 0.49 and 53.73 μM towards BDT and $E_2 17\beta G$, respectively, which are significantly lower than those of the wild type (1.96 and 149.9 μM). However, further $E_2 17\beta G$ transport assays revealed a decreased activity (Fig. 4g), which is consistent with the pathogenicity of R1150H. We also performed ATPase activity and $E_2 17\beta G$ transport activity assays with three additional variants (R1079A, R1083A and R1146A) which should also weaken the binding of the C-segment to TMD2. Similar to R1150H, all variants showed a decreased transport activity, and a lower $EC_{50}$ value compared to the wild type (Supplementary Fig. 10). These results further proved the essentiality of this C-segment binding conformation.

## Discussion

The R domain is an insertion found at various positions of some ABC proteins, previously named after its putative regulatory role[38]. In the cases of ABCA1[39] and ABCA4[40], the R domain succeeding the NBD interacts with its counterpart, presumably facilitating the dimerization of NBDs. An R domain succeeding the NBD was also found in the structure of ABCD1[41]. In contrast, the R domain, either at the N-terminus of ABCB1 homolog in *C. elegans*[42], or between the two modules of ABCB11[43], ABCC1 homolog[44] and ABCC7[27], was found occupying the translocation cavity. The previous reports found that the activities of yeast Ycf1 (ABCC1 homolog)[44,45] and human ABCC7[27,46] are regulated by the phosphorylation/dephosphorylation of the R domain. A recent structure of yeast Ycf1 showed that the R domain undergoes a conformational change upon dephosphorylation[45]. We noticed that the R domain of ABCC2 is also serine-rich (Supplementary Fig. 8g). To investigate the phosphorylation profile of the R domain of ABCC2, we performed mass spectrometry and it revealed that three residues (Ser878, Ser926 and Ser930) have a tendency to be phosphorylated (Supplementary Fig. 11a). However, all these three residues are missing in our structures due to the flexibility of the R domain. To mimic the phosphorylated state, we introduced triple mutations of S878D/S926D/S930D for the ATPase activity assays. It showed that the triple-mutant has a higher activity compared to the wild type (Supplementary Fig. 11b), but comparable to that of the mutant E892Q or E893Q (Fig. 2e), indicating that phosphorylation of these residues also abolishes the auto-inhibition of the R domain.

Based on the present structures, in which the R domain adopts various configurations at different transport states (Fig. 5), we proposed a regulatory role of R domain on the transport activity of ABCC2. At the rest state, as shown by the apo-form ABCC2 structure, the M-segment of R domain binds to the translocation cavity. Of note, in the trials of ABCC2 structure determination, we solved two other

apo-form structures at 3.6 and 4.2 Å, respectively (Supplementary Figs. 2, 6). One is from the same dataset of the aforementioned apo form, but with a poorer density of the R domain (termed apo', 3.6 Å) (Supplementary Fig. 3b), whereas the other is classified from the dataset of ATP/ADP-bound ABCC2 structure (termed apo'', 4.2 Å). All three apo-form structures adopt an IF conformation but differ from each other in the distance between the two NBDs (Supplementary Fig. 12), reflecting the process of NBD dimerization. Moreover, these three ABCC2 apo-form structures are reminiscent of the three apo states detected by bABCC1 single-molecule fluorescence resonance energy transfer (smFRET) assays, which showed three lowest smFRET states in the absence of ligand. Therefore, we have observed three bona fide lowest states of the apo form ABCC2 in the structures.

Upon BDT binding, the M-segment is expelled and released from the substrate-binding pocket, as shown in the BDT-bound ABCC2 structure (Figs. 3a, b and 5). As a member of MRP, ABCC2 can transport a wide spectrum of substrates, including endogenous metabolites and exogenous drugs[47]; thus, its activity needs to be precisely controlled. Based on our structural and biochemical analysis, it is plausible the R domain works as a competitive affinity filter, excluding the binding of some substrates under a certain affinity threshold, eventually ensuring the priority to export the conjugated bilirubin over the low-affinity substrates.

As deduced from our recently solved structure of ABCC4[48], ATP binding should further facilitate the transform of ABCC2 to an occluded conformation. Following the dimerization of NBDs, the ATP molecule at the consensus site is hydrolyzed, making ABCC2 adopt an OF conformation, and enabling substrate released to the bile duct, i.e., the active turnover state observed in our structure of ATP/ADP-bound ABCC2 (Figs. 4a and 5). In this conformation, the R domain functions as staples that crosslink TM14 and TM15 and freeze ABCC2 at the active turnover state. Moreover, this active turnover state of ABCC2 can perfectly interpret the molecular basis of the disease-related variant R1150H, and provides the structural evidence for the rate-limiting step in transport cycle as detected in the smFRET assays of bABCC1[35].

In summary, our study provides not only structural insights into the substrate specificity of conjugated bilirubin of ABCC2, but also the transport mechanism regulated by the R domain. Moreover, these findings shed light on the molecular basis of pathogenesis of the Dubin-Johnson disease.

## Methods
### Protein expression and purification
The codon-optimized gene encoding full-length human ABCC2 (Uni-prot: Q92887) was synthesized (Sangon Biotech, Shanghai) and cloned into a modified pCAG vector harboring a C-terminal Flag tag (DYKDDDDK) and an 8 × His tag using a ClonExpress® II One Step Cloning Kit (C113-02, Vazyme Biotech co., Ltd). Site-directed mutagenesis was performed using a standard two-step PCR, followed by purification using StarPrep Gel Extraction Kit (D205-04, GenStar, Beijing) and verified by DNA sequencing (Sangon Biotech, Shanghai).

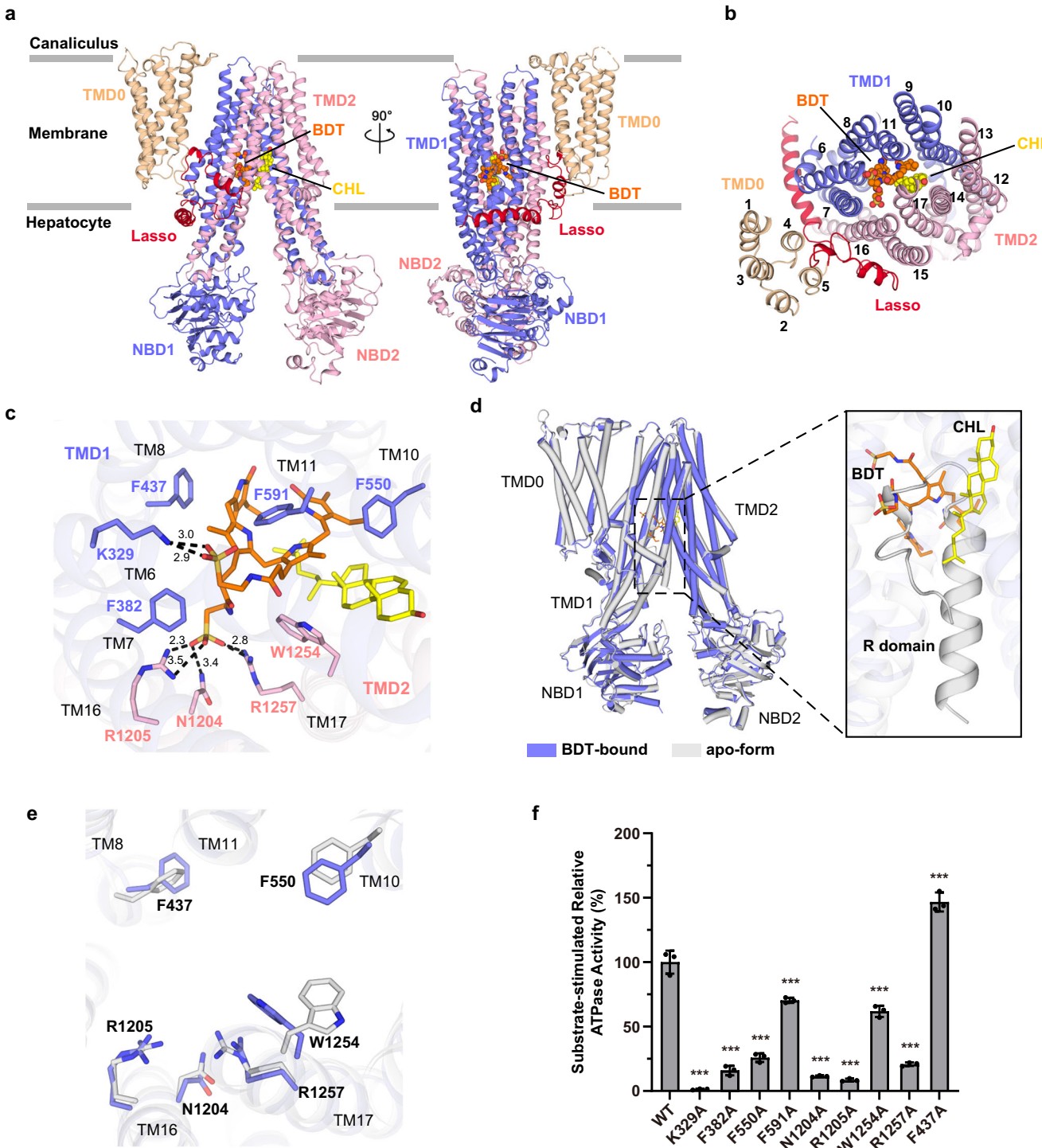

**Fig. 3 | Structure of BDT-bound ABCC2 and the substrate-binding pocket. a** Side and (**b**) top view of the overall structure of BDT-bound ABCC2 colored using the same color scheme as shown in Fig. 2a. The BDT and cholesterol (CHL) molecules are shown as orange and yellow spheres, respectively. **c** Zoom-in view of the BDT binding site. The BDT molecule, CHL molecule and the interacting residues of ABCC2 are shown as sticks. Carbon atoms are colored consistent with domain colors in (**a**), with oxygen in red, nitrogen in blue and sulfur in yellow. Hydrogen bonds and salt bridges are shown as black dotted lines, with the distances in Å. **d** Superposition of the apo-form against the BDT-bound ABCC2. The zoom-in image on the right shows the superposition of the R domain and BDT/CHL

molecules. **e** Superposition of the shared residues interacting with the R domain and BDT/CHL molecules from the apo-form (gray) and BDT-bound structure (slate), respectively. **f** Substrate-stimulated ATPase activities of ABCC2 WT and mutants that harboring a single mutation of residues at the substrate-binding pocket. Each data point is the average of independent experiments ($n = 3$), and error bars represent the means ± SD. One-way ANOVA is used for the comparison of statistical significance of WT and mutants. The $p$ values of all mutants are <0.0001. The $P$ values of <0.05, 0.01, and 0.001 are indicated with *, **, and ***, respectively. Source data are provided as a Source Data file.

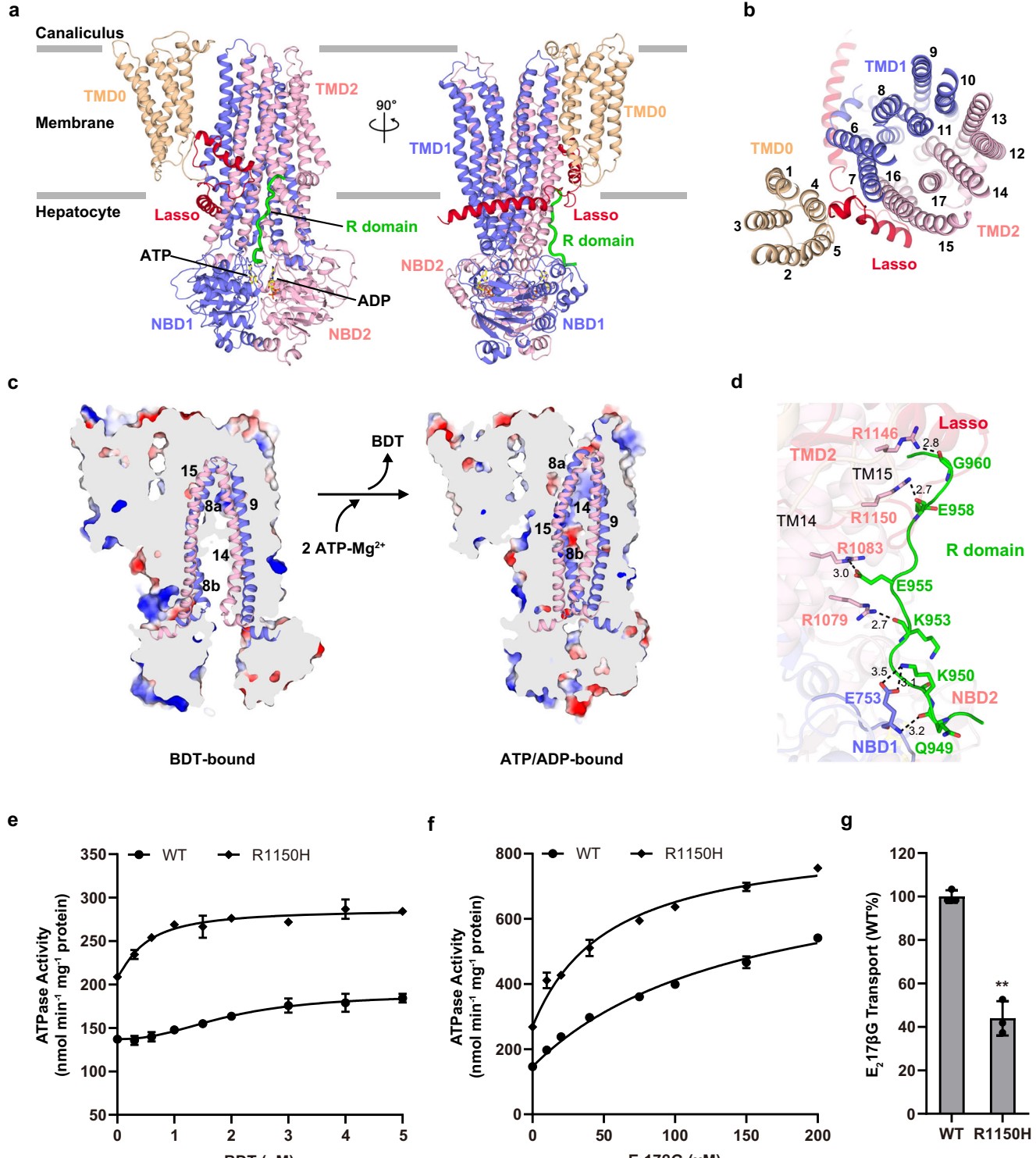

**Fig. 4 | Structure of ATP/ADP-bound ABCC2 and biochemical analysis of R domain anchored on TMD2. a** Side and (**b**) top view of the overall structure of ATP/ADP-bound ABCC2 with the same color scheme as in Fig. 2a. The ATP and ADP molecules are shown as yellow sticks, and Mg$^{2+}$ is shown as a green sphere. **c** Conformational changes upon ATP binding, shown as a cutaway representation of the electrostatic surface. TM8/TM9/TM14/TM15 are shown as cartoon. The electrostatic surface was generated by PyMOL 2.5.2 (https://pymol.org). **d** A detailed view of interface between the R domain and TMD2/NBD1. All interacting residues are shown as sticks. Hydrogen bonds and salt bridges are shown as black dotted lines, with the distances in Å. **e** The BDT- and (**f**) E$_2$17βG-stimulated ATPase activity

assays of ABCC2 and the R1150H mutant. The data points are fitted with the Hill equation in (**e**) and the Michaelis-Menten equation in (**f**). All data points for (**e**) and (**f**) represent means of three independent measurements (*n* = 3), and error bars represent the means ± SD. **g** The transport activity assays of ABCC2 the R1150H mutant using radioisotope-labeled substrate E$_2$17βG. The transport activities of mutants are normalized by WT. Each data point is the average of independent experiments (*n* = 3), and error bars represent the means ± SD. One-way ANOVA is used for the comparison of statistical significance. The *p* value of R1150H is 0.0085. The *P* values of <0.05, 0.01, and 0.001 are indicated with *, **, and ***, respectively. Source data are provided as a Source Data file.

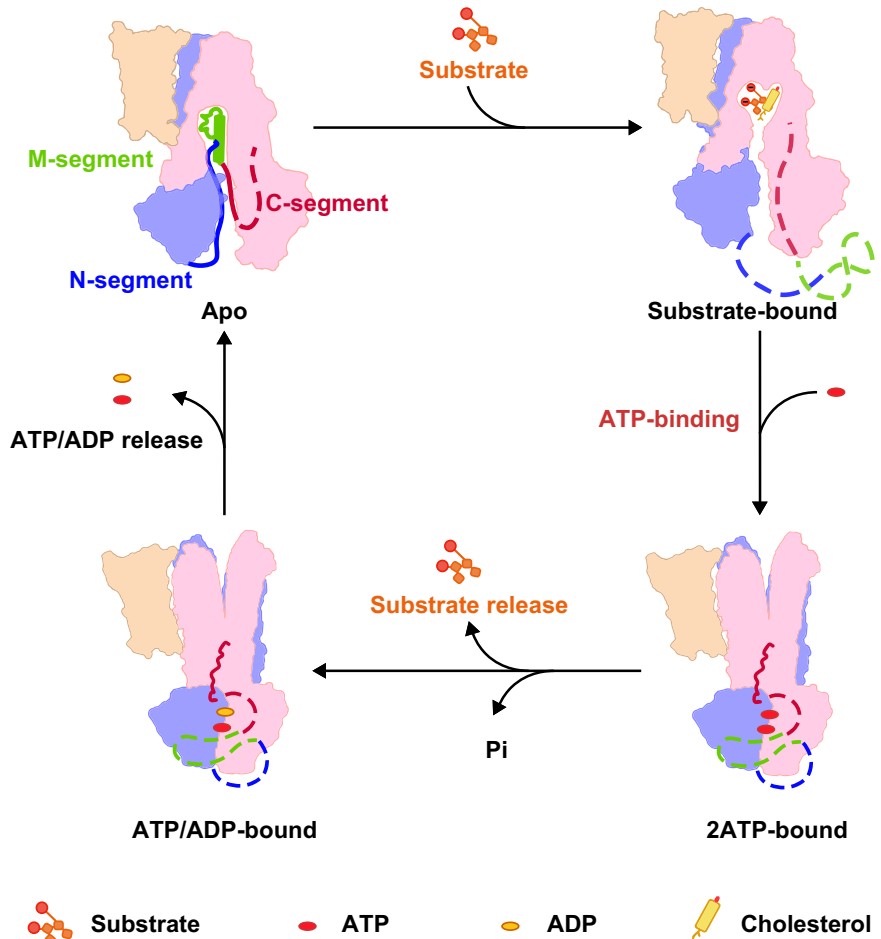

**Fig. 5 | A proposed model of the transport cycle driven by ABCC2.** A schematic illustration of the transport cycle of ABCC2. The apo-form ABCC2 adopts an inward-facing conformation, with M-segment of the R domain in the substrate-binding pocket. The N-segment is identified as a loop in the apo form, whereas the C-segment is missing. The M-segment is expelled upon substrate binding, resulting in the missing of R domain in this structure. The ATP binding and hydrolysis at consensus site facilitate the substrate release, meanwhile, the C-segment of the R domain docks to the lateral of TMD2, which might maintain this active turnover state. Finally, the release of nucleotides accompanied with dissociation of the NBD dimer resets ABCC2 to the rest state. TMD0 and the two half modules of TMD1&NBD1 and TMD2&NBD2 are colored with the same color scheme as Fig. 2a. The N-segment of the R domain is colored in blue, the M-segment is colored in green and the C-segment is colored in red, respectively. The untraceable regions of the R domain are represented by dashed lines.

The primers were synthesized by Sangon Biotech, and the sequences are listed in Supplementary information.

For protein expression, 800 mL HEK293F cells (R79007, Invitrogen) were cultured in SMM 293T-II medium (M293TII, Sino Biological Inc.) in a shaker, at 37 °C, under 5% CO$_2$. When the cell density reached ~2.5 × 10$^6$ cells per mL, ~1.8 mg plasmids and 4 mg PEIs (40816ES03; Yeasen, Shanghai, China) were pre-incubated in 45 ml fresh medium for 15–30 min and added to the cell culture with the supplement of another 100 ml fresh medium. Followed by 15-min static incubation, the transfected cells were cultured at 37 °C in 5% CO$_2$ for 48–60 h prior to harvest. After centrifuged at 4000 × $g$ for 5 min, cells were resuspended in 25 mM Tris-HCl pH 7.4, 150 mM NaCl. The suspension was frozen by liquid nitrogen and stored at −80 °C for further use.

For purification of ABCC2 for cryo-EM, all steps were performed at 4 °C. To extract ABCC2 form cell membrane, cells were rotated gently for 2 h in 25 mM Tris-HCl pH 7.4, 150 mM NaCl, 20% (v/v) glycerol, 1% (w/v) n-Dodecyl-β-D-Maltopyranoside (DDM, Bluepus) and 0.2% (w/v) cholesteryl hemisuccinate (CHS, Anatrace). Exceptionally, additional glibenclamide (G832360, Macklin) was added to cell membrane prior to membrane solubilization at a final concentration of 50 μM to purified ABCC2 for ATP-bound ABCC2 structure determination. After centrifugation at 45,000 rpm for 40 min, the supernatant was collected and incubated with the anti-FLAG M2 affinity gel (Sigma) for 1 h.

The resin was rinsed by buffer A containing 25 mM Tris-HCl pH 7.4, 150 mM NaCl, 10% glycerol (v/v), 0.06% digitonin (w/v) (BID3301, Apollo Scientific). Protein was eluted with buffer B containing 25 mM Tris-HCl pH 7.4, 150 mM NaCl, 5% glycerol (v/v), 0.06% digitonin (w/v) (BID3301, Apollo Scientific) supplemented with 200 μg/ml FLAG peptide. The protein elution was concentrated with a 100-kDa cut-off concentrator (Millipore) and loaded onto a size-exclusion column (Superose 6 Increase 10/300 GL, Cytiva) equilibrated with buffer C containing 25 mM Tris-HCl pH 7.4, 150 mM NaCl, 1 mM DTT, 0.06% digitonin (w/v). Peak fractions containing ABCC2 were pooled and concentrated for further biochemical studies or cryo-EM experiments.

All mutants used for biochemical assays and structure determination were expressed and purified in the same way as the wild-type protein.

### ATPase activity assays
The ATPase activities of ABCC2 and mutants were measured by quantitating inorganic phosphate using a modified malachite green-ammonium molybdate method based on previous described procedure[49].

To measure the ATPase activities of ABCC2 against different bilirubin ditaurate (BDT, disodium salt) (GC42931-10, GLPBIO) or estradiol-17β-D-glucuronide (E$_2$17βG, disodium salt) (GC10964-10,

GLPBIO) concentrations or varying ATP concentrations, protein at a final concentration of 0.05 μM was added to 75 μL reaction buffer containing 50 mM Tris-HCl, pH 7.4, 50 mM KCl, 1 mM DTT, 0.06% (w/v) digitonin, 2 mM MgCl$_2$. Then, BDT or E$_2$17βG was diluted into different concentrations and added into the reaction mixture. The reaction was started by the addition of ATP at a final concentration of 2 mM (for ATPase activity assay of ABCC2 against different substrates) or a varying ATP concentration and incubated for 30 min at 37 °C. Enzymatic reactions were terminated by addition of 25 μL the malachite development solution and incubated for 2 min at room temperature to form stable chromophore prior to adding 10 μL 10% oxalic acid. After 30 min, the 96-well plates were read at a wavelength of 630 nm on a SpectraMax iD5 Multi-Mode Microplate Reader (Molecular Devices). The released phosphate was quantified according to an absorbance standard curve established using KH$_2$PO$_4$ standards and the statistical analysis was performed using Origin 2021b (Academic). The malachite green development solution was prepared fresh before each experiment by mixing 10 mL of malachite green dye stock solution with 2.5 mL of 7.5% ammonium molybdate and 0.2 mL Tween 20 (Sigma-Aldrich).

## Cryo-EM sample preparation and data collection

To prepare the apo-form ABCC2 sample, 3.5 μL of purified ABCC2 at a concentration of ~9 mg/ml was applied to freshly glow-discharged QUANTIFOIL (R1.2/1.3, 300 mesh, holey carbon films) copper grids. The grids were blotted with filter paper for 5.0 s and zero blotting force. Then, the grids were plunged into liquid ethane cooled by liquid nitrogen using a Vitrobot Mark IV (FEI) under 100% humidity at 8 °C. A total of 2832 micrograph stacks were automatically collected with EPU 2 software[50] on a Titan Krios 300 kV transmission electron microscope equipped with a K3 Summit direct electron detector (Gatan) and a GIF Quantum energy filter (Gatan) at a defocus range of −2.0 to −1.2 μm with a magnification of ×81,000, resulting in a pixel size of 1.07 Å. Each movie stack containing 32 frames was exposed in a super-resolution mode, with a total dose of 55 e$^-$/Å$^2$.

To prepare BDT-bound ABCC2 complex sample, the purified ABCC2 concentrated to ~8 mg/ml was incubated with 593 μM BDT on ice for 30 min. After that, aliquots of 3.5 μL protein complex were applied to glow-discharged QUANTIFOIL (R1.2/1.3, 300 mesh, holey carbon films) Cu grids. The grids were blotted with filter paper with a 5 s blotting time and zero blotting force. Then the grids were plunged into liquid ethane cooled with liquid nitrogen using a Vitrobot Mark IV (FEI) under 100% humidity at 8 °C. Two datasets with a total of 1902 micrograph stacks were collected in the same manner as apo-form ABCC2.

To prepare ATP-bound ABCC2 complex sample, the purified ABCC2 was concentrated to ~6.6 mg/ml and mixed with 50 μM glibenclamide (G832360, Macklin). Followed by 30-min static incubation on ice, 10 mM ATP and 10 mM MgCl$_2$ were added to the protein mixture and incubated for another 15 min. After that, 3.5 μL of this sample was applied to glow-discharged QUANTIFOIL (R1.2/1.3, 300 mesh, holey carbon films) Cu grids, blotted for 5 s and zero blotting force. Then the grids were plunged into liquid ethane cooled with liquid nitrogen using a Vitrobot Mark IV (FEI) under 100% humidity at 8 °C. Two datasets with a total of 2250 micrograph stacks were collected with EPU 2 software on a Titan Krios microscope at 300 kV equipped with a K3 detector (Gatan) and a GIF Quantum energy filter (Gatan), at a magnification of ×81,000 with defocus values from −2.0 to −1.2 μm. For these stacks, motion correction and dose weighting were performed with patch motion correction with a Fourier cropping factor of 0.5, resulting in a pixel size of 1.07 Å. Meanwhile, the defocus values were estimated using Patch CTF estimation.

## Cryo-EM data processing

For the apo-form ABCC2 datasets, 3,449,154 automatically picked particles were extracted from the 2832 micrographs and subjected to 2D classification, using cryoSPARC v3.1.0[51]. 419,303 particles from good classes were used to generate templates for template picking against the entire dataset and a total of 2,412,788 particles were reselected and extracted. After 2D classification, a total of 625,168 particles were used for two rounds of ab initio reconstruction and heterogeneous refinement. After 3D classification, two types of inward-facing conformations with variable distance between the two NBDs were identified (the different opening angles for different classes are visualized in Supplementary Fig. 12b). We used two types of particles obtained from the heterogeneous refinement to perform homogeneous refinement followed by non-uniform refinement and local refinement in cryoSPARC[51]. A reconstruction map at 3.6 Å with a wider opening to the intracellular side was yielded by 175,142 apo' particles and its R domain cannot be confidential assigned due to poor density in corresponding region. The R domain density was observed more clearly between TMD1 and TMD2 in 170,268 apo particles, which has a narrower opening to the intracellular side than apo' particles, and the reconstruction map was obtained at a global resolution of 3.6 Å.

For BDT-bound ABCC2 datasets, A total of 565,668 particles were selected and extracted from two datasets to perform ab initio reconstruction and heterogeneous refinement, succeeding to blob picking, template picking and 2D classification. Ultimately, 368,404 particles of the best resolved class were selected for further refinements and yielded a reconstruction map at a global resolution of 3.3 Å.

For ATP-bound ABCC2 datasets, A total of 895,145 particles were selected from two datasets to perform four rounds of ab initio reconstruction and heterogeneous refinement, after blob picking, template picking and 2D classification. After 3D classification, two types of ABCC2 conformations were identified, including an outward-facing conformation that NBD dimer remains closed and an inward-facing conformation with a close opening to the intracellular side. We used these two types of particles to perform homogeneous refinement followed by non-uniform refinement and local refinement in cryoSPARC[51]. 131,148 outward-facing particles resulted in a 3.6 Å map, which showed ATP-Mg$^{2+}$ bound at degenerate site and an ADP molecule bound at consensus site. 79,970 inward-facing apo'' particles yielded a reconstruction map at global resolution of 4.2 Å.

## Model building and refinement

The structure model of apo-form ABCC2 was built based on an initial model predicted by AlphaFold2[52] and refined with secondary structure and geometry restraints, automatically or manually using Real-space refinement in PHENIX 1.20.1[53] or WinCoot 0.9.8.1[54]. Finally, residues Met1-Gln258, Val305-Ala856, Asp884-Ser914, Lys963-Ile1537 for apo-form were built in the map. For BDT-bound ABCC2, residue Met1-Glu89, Thr95-Gln258, Ser308-Gly867, Lys963-Ile1537 was built and refined according to the structure model of apo-form ABCC2 with a BDT molecule and a cholesterol molecule manually built into the extra densities between TMD1 and TMD2. Due to their relatively poor densities, the NBDs of apo- and BDT-bound ABCC2 were manually fitted into the map using the NBDs structure from ATP-bound ABCC2, and was automatically refined in PHENIX 1.20.1[53]. For the nucleotide-bound ABCC2, an outward-facing model of ABCC2 was generated by the SWISS-MODEL server[55], using the cryo-EM structure of bovine ABCC1 (PDB code 6UY0) as the reference model. After refined manually or automatically, Residues Met1-Pro194 for TMD0, Ser195-Lys263 for lasso motif, Ser308-Gly619, Lys963-Val1278 for TMDs, Asp620-Leu863, Glu1279-Ile1537 for NBDs and Lys947-Lys961 for R domain were built in the map with ATP-Mg$^{2+}$ and ADP fitted into prominent densities between two NBDs.

All structures were validated by PHENIX 1.20.1[53] and MolProbity 4.02[56]. The model refinement and validation statistics were

summarized in Supplementary Table 1. The UCSF ChimeraX 1.2.5[57] and PyMOL 2.5.2 (https://pymol.org) were used for preparing the structural figures. Protein sequences were aligned using Multalin (http://multalin.toulouse.inra.fr/multalin/) and the sequence-alignment figures were generated by ESPript 3 server (https://espript.ibcp.fr/).

## Membrane vesicle preparation
Inside-out membrane vesicles from HEK293F cells expressing human ABCC2 were prepared as described previously[43]. Briefly, HEK293F cells were collected at $3000 \times g$ for 10 min. Then, cell pellets were resuspended in 50 mM mannitol, 2 mM EGTA, 50 mM Tris-HCl, pH 7.0, supplemented with the protease inhibitor Cocktail (C0001, TargetMol, USA) and homogenized with a glass-Teflon tissue homogenizer. The homogenate was further centrifuged ($3000 \times g$, 10 min) to eliminate the unground pellets and large organelle. The supernatant was further centrifuged at $100,000 \times g$ for 60 min. Then, the sediment was resuspended in 50 mM sucrose, 100 mM $KNO_3$, 10 mM $Mg(NO_3)_2$, 10 mM Tris-HCl, pH 7.4. Finally, the vesicles were plunged into liquid nitrogen and stored at −80 °C for further use.

## Transport activity assays
Uptake of estradiol 17-(β-D-glucuronide) (E$_2$17βG) into membrane vesicles was performed following the rapid filtration method as described previously[58]. Shortly, the frozen vesicles were thawed at 37 °C and then incubated on ice in 10 mM Tris-HCl, pH 7.4, 50 mM sucrose, 100 mM $KNO_3$, 10 mM $Mg(NO_3)_2$, and 0.16 μM of [$^3$H]-estradiol 17-(β-D-glucuronide). The reaction was started by adding 2 mM ATP to the reaction mixture and for control experiments, 2 mM ADP instead of ATP was added. After incubation for 60 sec at 37 °C, E$_2$17βG uptakes were terminated by addition of 80 μL ice-cold stop buffer consisting of 10 mM Tris-HCl, pH 7.4, 50 mM sucrose, 100 mM $KNO_3$. The reaction mixture was filtrated by glass microfiber filters (GF/F, Whatman) with a nominal pore size of 0.7 μm. Then, the glass microfiber filters were washed by additional 6 mL stop buffer to eliminate excessive E$_2$17βG. The vesicle-associated radioactivity was determined by liquid scintillation counter. Data are presented as the means ± S.D. by biological repeats from three independent assays ($n = 3$).

## Mass spectrometry analysis of ABCC2 phosphorylation
The purified ABCC2 (0.94 mg/mL) was mixed with $5 \times$ SDS loading buffer. In each well of a 15% SDS-PAGE, 20 μL mixture was loaded for gel electrophoresis at 250 V for 75 min. Afterwards, ABCC2 protein band was extracted from the gel and treated with trypsin digestion. Then the peptides were analyzed by liquid chromatography coupled to tandem MS (LC-MS/MS) (Thermofisher Q Exactive Plus). The LC system was equipped with an analytical column (Acclaim PepMap RSLC, 50 μm × 15 cm, nanoViper, C18, 2 μm particle, 100 Å, Thermo Fisher Scientific). Mobile phases A (0.1% formic acid) and B (0.1% formic acid, 80% acetonitrile) were applied to separate the peptides. The flow rate was set at 300 nL min$^{-1}$. The numbers of phosphorylated fragments were identified by LC−MS/MS data.

## Reporting summary
Further information on research design is available in the Nature Portfolio Reporting Summary linked to this article.

## Data availability
The data that support this study are available from the corresponding authors upon request. The cryo-EM density maps of five structures have been deposited at the Electron Microscopy Data Bank under accession codes EMD-36691 (apo-form ABCC2), EMD-36709 (BDT-bound ABCC2), EMD-36713 (ATP/ADP-bound ABCC2), EMD-36719 (apo′-form ABCC2), and EMD-36720 (apo″-form ABCC2). and coordinates have been deposited in the Protein Data Bank (PDB) under accession codes: 8JX7 (apo-form ABCC2), 8JXQ (BDT-bound ABCC2), 8JXU (ATP/ ADP-bound ABCC2), 8JY4 (apo′-form ABCC2) and 8JY5 (apo″-form ABCC2). Source data are provided with this paper.

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

## Acknowledgements

We thank the Cryo-EM Center at University of Science and Technology of China for the support of cryo-EM data collection. We thank the Experiment Center for Life Science at University of Science and Technology of China for the support of experimental techniques. This work was supported by the Strategic Priority Research Program of the Chinese Academy of Sciences (XDB37020202), the Ministry of Science and Technology of China (2019YFA0508500), National Natural Science Foundation of China (32371257), Research Funds of Center for Advanced Interdisciplinary Science and Biomedicine of IHM (QYZD20220001) and China Postdoctoral Science Foundation (2022M723050).

## Author contributions

Y.C. and W.-T.H. conceived the project and planned the experiments. Y.-X.M. and Z.-P.C. expressed and purified human ABCC2. Y.-X.M., Z.-P.C. and L.W. performed cryo-EM data collection, structure determination

and model refinement. Y.-X.M. and J.W. performed biochemistry experiments. W.-T.H., Z.-P.C., Y.C. and C.-Z.Z. wrote the manuscript. All authors read and edited the paper.

## Competing interests

The authors declare no competing interests.
