## [Peer Review File · Nature Communications]

Transport mechanism of human bilirubin transporter ABCC2 tuned by the inter-module regulatory domainReviewer #1 (Remarks to the Author):

Authors of the manuscript "Transport mechanism of human bilirubin transporter ABCC2 tuned by the inter-module regulatory domain", present exciting results on structural and functional analysis of a human ABC transporter of C family. ABCC2 is a bilirubin transporter and has implications on human health when dysfunctional. ABCC family members of ABC transporters have a R-domain and recently it has been shown to have a regulatory role in the transport cycle in multiple publications. Mao et. al., here, have used cryoEM and biochemical assays as tools to elaborate on this mechanism. They show three cryoEM structures representative of three different steps of the transport cycle and demonstrate the role of R-domain in the regulation of transport.

There are few concerns on the manuscript that should be addressed by the authors.

1. In ABCC family transporters, regulation by regulatory domain (R-domain) is mainly controlled by phosphorylation and dephosphorylation. Previous reports on structural analysis of ABCC family members have been shown that phosphorylated state is active and dephosphorylated state is inactive. Following publications have emphasized on the role of phosphorylation and exhibited how ATPase activity and hence transport is dependent on the phosphorylation of R-domain- Liu et. al., 2017, Cell <http://dx.doi.org/10.1016/j.cell.2017.02.024>; Zhang et. al., 2018, PNAS <https://doi.org/10.1073/pnas.1815287115>; Zhang & Chen, 2016, Cell <http://dx.doi.org/10.1016/j.cell.2016.11.014> ; Zhang et. al., 2017, Cell <http://dx.doi.org/10.1016/j.cell.2017.06.041>; Bickers et. al. 2021, PNAS <https://doi.org/10.1073/pnas.2025853118>; Khandelwal et. al., 2022, Nat. Commun. <https://doi.org/10.1038/s41467-022-28811-w>. The sites of phosphorylation are shown to be fairly conserved. Interestingly, a previous publication on CFTR (Zhang et. al., 2017, Cell <http://dx.doi.org/10.1016/j.cell.2017.06.041>) and a recent BioRxiv preprint on a ABCC family transporter Ycf1 (Khandelwal et. al., BioRxiv, 2023, <https://doi.org/10.1101/2023.06.22.546176>.) have shown dephosphorylation dependent movement of R-domain in between NBDs and into the substrate binding cavity, respectively. It is similar to what authors have demonstrated here. It will be interesting if authors discuss their findings in the light of these other published reports.

Since phosphorylation is extremely central to the process of this auto-regulation by R-domain as demonstrated in other publications on ABCC family members, authors should show the status of phosphorylation of R-domain in their purified protein and if present, what residues are modified. What is the effect on ATPase activity in phosphorylated vs non-phosphorylated ABCC2?

2. Authors performed transport assays with E217 β G as a substrate while in the substrate-bound structure they use BDT. Were there any limitations in using the same substrate for transport assays as the structures? Please elaborate on this.

3. In the substrate-bound structure, a cholesterol molecule is present with a BDT molecule in the substrate binding cavity. It is an interesting observation and can be further discussed. Is cholesterol source endogenous or is it CHS from the detergent? Is it possible cholesterol is transported with BDT? Why is there no cholesterol in the Apo structures?

4. The manuscript says three structures but table 1 and supplementary data show a total of five structures. There are multiple ABCC2 Apo structures reported here. Are there any differences in these Apo structures? Please clarify it in the text.

5. The clash score is really high in all the models here specially for the maps of reported resolution. Authors should closely inspect it and try to fix these clashes.

6. Please show the measured distances in Fig 2d, 3c, 4d for the interactions mentioned.

7. In Fig 3f, the statistical significance representation on the graph is confusing. Please make it clearer for the audience.

8. F437 of TMD 1 appears to be important for the BDT interaction in the substrate binding pocket via π - π interactions with the pyrroles. In Fig 3f, why F437A mutant shows a higher activity than the WT in the presence of the substrate even though it is one of the shared residues with the R-domain and substrate interaction?

9. Please provide the source data for gel, ATPase assays, maps, models associated with the manuscript. It will be useful for the review process.

Reviewer #2 (Remarks to the Author):

The authors report several structures of human ABCC2/MRP2, an ABC transporter that holds significant physiological relevance as a multidrug exporter and transporter for bilirubin whose dysfunction is linked to liver disease. While these structures, which are reported in apo, substrate bound, and nucleotide bound states, are important on their own, a key finding is the role of the regulatory domain located in the linker between the two halves of the transporter in different states of the substrate transport cycle. In the apo state, the authors resolve structural elements from the RD within the substrate binding cavity, which are dislodged upon substrate binding. The authors postulate that the RD serves as an 'affinity filter' allowing passage of higher affinity substrates that can dislodge it. IN the ATP bound state, the RD is seen to make contacts with TMD2, and the authors postulate it plays a role in stabilizing the outward facing state, promoting substrate release. Functional analysis in the form of substrate stimulated ATPase activity assays and radioactive substrate transport in membrane vesicles are performed for the wildtype protein and several substrate binding site mutants as well as mutants targeting the RD interaction with TMD2 in the ATP bound state. Based on their results, the authors put forward a mechanistic model for ABCC2 function emphasizing the role of the RD. The manuscript is, in principle, suitable for publication in Nature communications based on the presentation of the first structures of human ABCC2 and important mechanistic insights gained from the combined structure-function studies. However, the following points should be considered final decision on publication is made:

Major Points:

1. The authors model in a cholesterol molecule in the substrate binding site of their BDT bound structure. What conclusive evidence do the authors have for cholesterol occupancy at this site? While it may well be there, could the density features also arise from positional disorder in substrate orientation (ie mixture of 2 or more orientations leading to averaged density)? IN the absence of supporting experimental evidence, the authors should add the caveat that cholesterol binding concurrently with the substrate is a speculation. What is the source of this cholesterol molecule considering no cholesterol supplementation was done in the digitonin solubilized samples, considering that in the absence of substrate the site is purportedly occupied by the RD?

2. The argument for the role of the RD in the outward facing conformation would be strengthened by analysis of additional mutants beyond R1150. Why were additional mutants not tested to see, at the very least, their effect on substrate transport and ATPase activity?

Minor Points:

3. Lines 46-49 mention the role of ABCC2 in transport of estradiol and read a bit awkwardly. Do the authors mean that ABCC2 activity leads to estradiol conjugate mediated inhibition of ABCB11 and causes cholestasis? Is the role of ABCC2 in development of cholestasis well established?

4. Lines 116-120, the discussion of TMD0 should distinguish between B and C family TMD0s more clearly.

Reviewer Comments:

Reviewer #1 (Remarks to the Author):

Authors of the manuscript “Transport mechanism of human bilirubin transporter ABCC2 tuned by the inter-module regulatory domain”, present exciting results on structural and functional analysis of a human ABC transporter of C family. ABCC2 is a bilirubin transporter and has implications on human health when dysfunctional. ABCC family members of ABC transporters have a R-domain and recently it has been shown to have a regulatory role in the transport cycle in multiple publications. Mao et. al., here, have used cryoEM and biochemical assays as tools to elaborate on this mechanism. They show three cryoEM structures representative of three different steps of the transport cycle and demonstrate the role of R-domain in the regulation of transport.

There are few concerns on the manuscript that should be addressed by the authors.

Q1. In ABCC family transporters, regulation by regulatory domain (R-domain) is mainly controlled by phosphorylation and dephosphorylation. Previous reports on structural analysis of ABCC family members have been shown that phosphorylated state is active and dephosphorylated state is inactive. Following publications have emphasized on the role of phosphorylation and exhibited how ATPase activity and hence transport is dependent on the phosphorylation of R-domain-

Liu et. al., 2017, Cell <http://dx.doi.org/10.1016/j.cell.2017.02.024>; Zhang et. al., 2018, PNAS <https://doi.org/10.1073/pnas.1815287115>; Zhang & Chen, 2016, Cell <http://dx.doi.org/10.1016/j.cell.2016.11.014>; Zhang et. al., 2017, Cell <http://dx.doi.org/10.1016/j.cell.2017.06.041>; Bickers et. al. 2021, PNAS <https://doi.org/10.1073/pnas.2025853118>; Khandelwal et. al., 2022, Nat. Commun. <https://doi.org/10.1038/s41467-022-28811-w>. The sites of phosphorylation are shown to be fairly conserved. Interestingly, a previous publication on CFTR (Zhang et. al., 2017, Cell <http://dx.doi.org/10.1016/j.cell.2017.06.041>) and a recent BioRxiv preprint on a ABCC family transporter Ycf1 (Khandelwal et. al., BioRxiv, 2023, <https://doi.org/10.1101/2023.06.22.546176>.) have shown dephosphorylation dependent movement of R-domain in between NBDs and into the substrate binding cavity, respectively. It is similar to what authors have demonstrated here.

It will be interesting if authors discuss their findings in the light of these other published reports.

A: Thank you for the suggestions. We have incorporated discussion of these findings in the Discussion section.

Q2: Since phosphorylation is extremely central to the process of this auto-regulation by R-domain as demonstrated in other publications on ABCC family members, authors should show the status of phosphorylation of R-domain in their purified protein and if present, what residues are modified. What is the effect on ATPase activity in phosphorylated vs non-phosphorylated ABCC2?

A: We have examined the status of phosphorylation of the R domain by mass spectrometry, and the result revealed that three residues (S878, S926 and S930) have a tendency to be phosphorylated (Supplementary Fig. 11a). However, none of these three residues could be assigned in our structures due to flexibility of the R domain. Afterwards, these three residues were mutated into aspartic acids to mimic the phosphorylated state. All these mutants showed higher ATPase activities compared to

the wild type (Supplementary Fig. 11b), but comparable to that of E892Q and E893Q (Fig. 2d, e). Notably, E892 and E893 are two residues on the R domain as we observed in the apo-form structure. It indicated that phosphorylation of these residues also abolishes the auto-inhibition of the R domain.

Q3. Authors performed transport assays with E₂17βG as a substrate while in the substrate-bound structure they use BDT. Were there any limitations in using the same substrate for transport assays as the structures? Please elaborate on this.

A: The radioactive labeled BDT is not commercially available. Thus, we used the radioactive E₂17βG for the transport activity assays.

Q4. In the substrate-bound structure, a cholesterol molecule is present with a BDT molecule in the substrate binding cavity. It is an interesting observation and can be further discussed. Is cholesterol source endogenous or is it CHS from the detergent? Is it possible cholesterol is transported with BDT? Why is there no cholesterol in the Apo structures?

A: Thank you very much for the comments. We have carefully checked the density, and it is well fitted with a cholesterol molecule, but not a CHS molecule, while the map was contoured at 5 σ (Supplementary Fig. 8b). The cholesterol, which is abundant in human cell membrane, is most likely co-purified with ABCC2.

Concerning that the cholesterol helps to stabilize BDT in the substrate-binding pocket, it suggested that the cholesterol is necessary for the transport of BDT, but not co-transported with BDT. The R domain occupies the corresponding binding pocket, thus the cholesterol is absent in the apo-form structure.

Q5. The manuscript says three structures but table 1 and supplementary data show a total of five structures. There are multiple ABCC2 Apo structures reported here. Are there any differences in these Apo structures? Please clarify it in the text.

A: We have actually obtained two more apo-form structures at a lower resolution, which are discussed in line 268-279 of revised manuscript, and shown in Supplementary Fig. 12. These apo-form structures mainly differ from each other in the flexibility of the R domain and the distance between the two NBDs. These apo-form structures represent the three bona fide lowest states of the apo form as detected in bABCC1 smFRET assays (<https://doi.org/10.7554/eLife.56451>). We have modified our statements in the text to avoid this confusion as you have suggested.

Q6. The clash score is really high in all the models here specially for the maps of reported resolution. Authors should closely inspect it and try to fix these clashes.

A: We have carefully refined all structures.

Q7. Please show the measured distances in Fig 2d, 3c, 4d for the interactions mentioned.

A: Thank you for the suggestion. The distances have been shown in the corresponding figures.

Q8. In Fig 3f, the statistical significance representation on the graph is confusing. Please make it clearer for the audience.

A: Revised.

Q9. F437 of TMD 1 appears to be important for the BDT interaction in the substrate binding pocket via π - π interactions with the pyrroles. In Fig 3f, why F437A mutant shows a higher activity than the WT in the presence of the substrate even though it is one of the shared residues with the R-domain and substrate interaction?

A: In fact, both the basal and the BDT-stimulated activities of F437A are lower than WT, as shown below. However, the relative BDT-stimulated activity to its basal activity is higher than that of WT, due to the sharp decrease of the basal activity upon the introduction of an F437A mutation (Supplementary Fig. 8c).

Q10. Please provide the source data for gel, ATPase assays, maps, models associated with the manuscript. It will be useful for the review process.

A: You can download the source data from the network disc through this link: <https://rec.ustc.edu.cn/share/3ab2fd90-8cd2-11ee-b098-4f382c419995> (password: yeh5).

Reviewer #2 (Remarks to the Author):

The authors report several structures of human ABCC2/MRP2, an ABC transporter that holds significant physiological relevance as a multidrug exporter and transporter for bilirubin whose dysfunction is linked to liver disease. While these structures, which are reported in apo, substrate bound, and nucleotide bound states, are important on their own, a key finding is the role of the regulatory domain located in the linker between the two halves of the transporter in different states of the substrate transport cycle. In the apo state, the authors resolve structural elements from the RD within the substrate binding cavity, which are dislodged upon substrate binding. The authors postulate that the RD serves as an ‘affinity filter’ allowing passage of higher affinity substrates that can dislodge it. IN the ATP bound state, the RD is seen to make contacts with TMD2, and the authors postulate it plays a role in stabilizing the outward facing state, promoting substrate release. Functional analysis in the form of substrate stimulated ATPase activity assays and radioactive substrate transport in membrane vesicles are performed for the wildtype protein and several substrate binding site mutants as well as mutants targeting the RD interaction with TMD2 in the ATP bound state. Based on their results, the authors put forward a mechanistic model for ABCC2 function emphasizing the role of the RD. The manuscript is, in principle, suitable for publication in Nature communications based on the presentation of the first structures of human ABCC2 and important mechanistic insights gained from the combined structure-function studies. However, the following points should be considered final decision on publication is made:

Major Points:

Q1. The authors model in a cholesterol molecule in the substrate binding site of their BDT bound structure. What conclusive evidence do the authors have for cholesterol occupancy at this site? While it may well be there, could the density features also arise from positional disorder in substrate orientation (ie mixture of 2 or more orientations leading to averaged density)? IN the absence of supporting experimental evidence, the authors should add the caveat that cholesterol binding concurrently with the substrate is a speculation. What is the source of this cholesterol molecule considering no cholesterol supplementation was done in the digitonin solubilized samples, considering that in the absence of substrate the site is purportedly occupied by the RD?

A: Thank you very much for the comment. We have carefully checked the density, and it is well

fitted with a cholesterol molecule, but not a CHS molecule or other detergents, while the map was contoured at 5 σ (Supplementary Fig. 8a, b). The cholesterol, which is abundant in human cell membrane, is most likely co-purified with ABCC2.

Concerning that the cholesterol helps to stabilize BDT in the substrate-binding pocket, it suggested that the cholesterol is necessary for the transport of BDT, but not co-transported with BDT. The R domain occupies the corresponding binding pocket, thus the cholesterol is absent in the apo-form structure.

According to your suggestion, we have added the caveat that cholesterol binding concurrently with the substrate is a speculation in the manuscript.

Q2. The argument for the role of the RD in the outward facing conformation would be strengthened by analysis of additional mutants beyond R1150. Why were additional mutants not tested to see, at the very least, their effect on substrate transport and ATPase activity?

A: Thank you for the suggestion. We have performed activity assays of additional mutants, including R1079A, R1083A and R1146A, which are involved with RD binding to TMD2. The assays showed similar results as R1150H, which have been supplemented in the revised manuscript as supplementary Fig. 10.

Minor Points:

Q3. Lines 46-49 mention the role of ABCC2 in transport of estradiol and read a bit awkwardly. Do the authors mean that ABCC2 activity leads to estradiol conjugate mediated inhibition of ABCB11 and causes cholestasis? Is the role of ABCC2 in development of cholestasis well established?

A: In fact, a couple of previous reports showed that ABCC2 contributing to the development of cholestasis in both *in vitro* and *in vivo* studies. For instance, one study demonstrated that E₂17 β G inhibited the transport activity of ABCB11 in canalicular plasma membrane vesicles, but not in ABCC2-deficient vesicles ([https://doi.org/10.1016/S0016-5085\(00\)70224-1](https://doi.org/10.1016/S0016-5085(00)70224-1)). Another study in rats revealed that ABCC2 is essential for E₂17 β G to induce cholestasis (<https://doi.org/10.1053/jhep.2000.8263>). However, a further study is needed for the role of ABCC2 in development of cholestasis in human. Following your advice, we have modified the statements in lines 50-55.

Q4. Lines 116-120, the discussion of TMD0 should distinguish between B and C family TMD0s more clearly.

A: Revised.

Reviewer #1 (Remarks to the Author):

I appreciate the authors responding to all my previous comments diligently. This manuscript will be a good addition to the overall research supporting the regulatory role of R domain in this family of ABC transporters.

Reviewer #2 (Remarks to the Author):

The authors have addressed all my points.